# Phasic Dopamine Changes and Hebbian Mechanisms during Probabilistic Reversal Learning in Striatal Circuits: A Computational Study

**DOI:** 10.3390/ijms23073452

**Published:** 2022-03-22

**Authors:** Miriam Schirru, Florence Véronneau-Veilleux, Fahima Nekka, Mauro Ursino

**Affiliations:** 1Department of Electrical, Electronic and Information Engineering Guglielmo Marconi, University of Bologna, Campus of Cesena, 47521 Cesena, Italy; miriam.schirru2@studio.unibo.it; 2Faculté de Pharmacie, Université de Montréal, Montreal, QC H3T 1J4, Canada; florencevv@msn.com (F.V.-V.); fahima.nekka@umontreal.ca (F.N.); 3Centre de Recherches Mathématiques, Université de Montréal, Montreal, QC H3T 1J4, Canada; 4Centre for Applied Mathematics in Bioscience and Medicine (CAMBAM), McGill University, Montreal, QC H3G 1Y6, Canada

**Keywords:** neurocomputational model, basal ganglia, orbitofrontal cortex, probabilistic reinforcement learning, reversal learning, synapse Hebbian training, behavioral flexibility, dopamine phasic changes

## Abstract

Cognitive flexibility is essential to modify our behavior in a non-stationary environment and is often explored by reversal learning tasks. The basal ganglia (BG) dopaminergic system, under a top-down control of the pre-frontal cortex, is known to be involved in flexible action selection through reinforcement learning. However, how adaptive dopamine changes regulate this process and learning mechanisms for training the striatal synapses remain open questions. The current study uses a neurocomputational model of the BG, based on dopamine-dependent direct (Go) and indirect (NoGo) pathways, to investigate reinforcement learning in a probabilistic environment through a task that associates different stimuli to different actions. Here, we investigated: the efficacy of several versions of the Hebb rule, based on covariance between pre- and post-synaptic neurons, as well as the required control in phasic dopamine changes crucial to achieving a proper reversal learning. Furthermore, an original mechanism for modulating the phasic dopamine changes is proposed, assuming that the expected reward probability is coded by the activity of the winner Go neuron before a reward/punishment takes place. Simulations show that this original formulation for an automatic phasic dopamine control allows the achievement of a good flexible reversal even in difficult conditions. The current outcomes may contribute to understanding the mechanisms for active control of dopamine changes during flexible behavior. In perspective, it may be applied in neuropsychiatric or neurological disorders, such as Parkinson’s or schizophrenia, in which reinforcement learning is impaired.

## 1. Introduction

Human beings have the capacity to adapt their choices in a very flexible way in different conditions, depending on the particular scenario in which an action occurs and on previous history of rewards and punishments; a situation described in the literature as reinforcement learning. In particular, in a non-stationary environment, individuals must be able to modify their behavior so that an action previously rewarded can be seen in another scenario as inappropriate and replaced with another choice. In recent years, a large number of studies employed reversal learning paradigms to assess individual variability in reward based tasks. Some of these studies were also aiming to achieve a deeper understanding of cognitive alterations in neurological and neuropsychiatric disorders, such as Parkinson’s disease, schizophrenia, depression, compulsive disorders, or substance abuse.

There is a large consensus in the literature that reversal learning involves at least two main brain structures, the basal ganglia (BG) and the prefrontal cortex, PFC (although amygdala and hippocampus are also involved in several tasks, see [1]). In particular, cortico-striatal synapses, which represent the initial projections of the BG processing stream linking the cortical structures to the striatum, experiment Hebbian plasticity in response to reward and punishments, a phenomenon mediated by dopamine phasic changes [2,3]. Neuroimaging studies illustrate the involvement of the dorsal [4] and ventral [5] regions of the striatum during reversal learning. Moreover, reversal learning is impaired in subjects with focal BG lesions [6] and in parkinsonian subjects [7,8]; it is affected by levodopa administration, a dopamine precursor [9] and by drug manipulation of D1 and D2 receptors [10,11,12].

On the other hand, several studies suggest that the BG is under the control of the PFC (especially the orbito-frontal cortex, OFC) and this influence is essential to correctly perform reversal learning. Neuroimaging studies show increased activity in the OFC in human subjects performing reversal learning [5,13,14], while deficits in reversal learning have been seen after lesions of the OFC [15,16]. Moreover, insufficient OFC functioning can determine impaired learning in various psychiatric disorders, such as schizophrenia [17] and obsessive compulsive disorders [18]. Furthermore, direct current stimulation over the ventrolateral PFC causes a greater number of perseverative errors, thus impairing reversal learning [19]. It is generally assumed that the PFC areas exert top-down control on the BG [20] and this control is especially important to suppress previously learned responses [14].

A fundamental role in this process is played by dopamine (DA) changes. A traditional point of view is that dopamine neurons, projecting to the striatum, code for reward prediction errors through phasic release, thus mediating information processing in the PFC and the BG [21,22,23]. On the other hand, recent studies [24,25,26,27,28] have found the presence of a sustained (tonic) DA response directed towards future rewards, arguing that it may represent a motivational signal. Various recent studies underline the fundamental link existing between flexible learning and dopamine [29,30]. Cools et al. [31] observed that the basal dopamine level, tonic, in the striatum is correlated with the way reversal learning is performed: subjects with high baseline dopamine exhibit relatively better reversal learning from unexpected rewards than from unexpected punishments, whereas subjects with low baseline dopamine showed the opposite behavior.

However, as declared by several authors even in very recent studies [32], “the precise relationship between dopamine and flexible decision making remains unclear”. Westbroock et al. [33] suggest that dopamine signaling in distinct cortico-striatal subregions represents a balance between costs and benefits, emphasizing the rich spatiotemporal dynamics of dopamine release in striatal subregions.

The neural mechanisms behind reversal learning in the PFC-BG circuitry and the role of dopamine can be better understood by the use of neurocomputational models, inspired by the biology. Indeed, many such models, based on the classic BG division in Go–NoGo pathways, have been presented in recent years, with important gain of knowledge. Frank developed a model for the BG-dopamine system to provide insights into the BG adaptive choices and the impairments in Parkinson disease (PD) subjects [34]. In order to overcome previous limitations, a more recent expanded version of their model [20] includes a module, simulating the OFC, which represents contextual information from previous experiences. With this model, the authors explored conditions when more complex decision-making tasks (including reversal learning) require a top-down control and demonstrate the role of the OFC in these conditions. Berthet et al. [35] implemented Hebbian Go and NoGo connectivity in a Bayesian network (in which units represent probabilities of stochastic events) and tested the model in several learning paradigms, including reversal learning. An important result of this study is that a flexible strategy for action choices is needed to solve different tasks, hence emphasizing the necessity for an external control. To simulate reversal learning, Moustafa et al. [36] extended a previous model of the BG, including the role of working memory in the PFC, thus implementing an actor-critic schema. Synapses were trained with Hebbian mechanisms, under the control of the critic by the PFC module, which calculates a temporal difference error signal to affect synapses. The models by Morita et al. [12,37,38] based on a state transition approach, try to understand how dopamine changes are controlled by upstream circuitry and suggest that synapse forgetting and an increase of the reward after dopamine depletion are an essential mechanism to explain the results of many effort-related tasks. Using a mechanistic model of the BG, Mulcahy et al. [39] observed that after learning, a system becomes biased to the previously rewarded choice; hence, establishing the new choice required a greater number of trials. Moreover, with this model, the authors were able to simulate conditions occurring in PD and Huntington’s diseases.

Although the previous experiments and modeling studies depict a clear basic scenario, some aspects deserve further analysis. In our opinion, most experiments in the literature concerning reversal learning, and most matching neurocomputational simulation results, are based on a quite unnatural paradigm. In the classic probabilistic reversal learning paradigm, it is assumed that one action is more frequently rewarded while another action is more frequently punished. The role of the two actions is then reverted, by exchanging the relative probabilities. However, in our daily life, the significance of actions is more complex than in this previous basic schema. Assuming that many actions can have a function in our life, rewards or punishments generally depend on the context in which an action is performed. An action which is more often rewarded in a given context (that is, in response to a given stimulus), can be rewarded less frequently and should be avoided in a different context (that is, when the input stimulus changes). Humans must establish a kind of hetero-associative relationship between actions and stimuli, and these associations can change in a non-stationary environment. In other terms, it is not possible to assign a value to an action without considering the particular input conditions (stimulus or context) on which an action must be performed. This poses more stringent constraints on the learning rules adopted, and makes the training and reversal learning more complex than in most previous basic experiments.

The aim of the present work is to reconsider the overall problem of probabilistic reversal learning, by investigating a multi-action task, using a biologically inspired model of the human BG we developed in previous years (see Appendix A for a detailed description of the model) [40,41]. In this study, we assume that all the selected actions are correct at least in one context, hence, should be selected by the agent when the correct stimulus is present. The task is not to choose only one correct action and refuse all the other actions, but to associate each stimulus to a different action in a random probabilistic environment. Examples are given first in the case of a two-action choice and then for a four-action choice.

As in previous models, we assume that synapses in the striatum are plastic and their changes are governed by Hebbian mechanisms. Moreover, we assume that rewards and punishments (which are now stimulus-dependent) are mimicked via phasic peaks and phasic drops of striatal dopamine.

Two important aspects that, to our best knowledge, were not discussed in previous works are assessed here. First, not all versions of the Hebb rule behave in the same way, assuming that the network starts from a completely naïve condition and must learn different stimulus-action associations. In particular, we show that a Hebb rule which changes synapses only with reference to the post-synaptic winner neuron outperforms the other ones. Second, reverse learning requires that the previous learned action is forgotten, to avoid excessive competition between alternative choices. This requires additional top-down mechanisms allowing a flexible use of dopamine, especially producing a stronger dopamine dip in case of unexpected punishments. The latter aspect is discussed, on the basis of a potential top-down flexible control.

These results are important to achieve a deeper understanding of possible learning rules in the BG and, above all, to develop new concepts on how dopamine could be flexibly modified by an action-critics mechanism, hence providing ideas on a possible role of the PFC. Finally, the present algorithm can find potential application in biologically inspired AI algorithms, devoted to learn flexible stimulus-action associations in a non-stationary probabilistic environment.

## 2. Results

### 2.1. Two-Action Choice

This section illustrates the results obtained during a test phase, after a phase of training. In the following, the term “agent” will be used to denote any of the 10 independent learning procedures, reported in detail in the Section 4. Simulation results are described for ten agents performing a two choices probabilistic task. All simulations are performed with a normal tonic dopaminergic input (*D* = 1.0), corresponding to healthy levels. During the training phase, the synapses are modified based on a previous history of rewards and punishments, whereas the test is implemented without phasic change in dopamine (neither rewards nor punishments). Phasic dopamine changes following rewards and punishments, affect the Go and NoGo activity, resulting in a variation of the synapses. In basal training, phasic peak during reward is twice the basal value, whereas the dopamine drop during punishment is half the tonic dopamine value. As it will be explained further on, reversal training conditions will require enhanced punishments.

All synapses subject to training are modified according to five different learning rules, as described in the Section 4. For each rule, we compute the number of correct responses for the first choice (with high probability of reward during training) and for the second choice (with small probability of reward during training). Each agent performs the test 100 times (50 actions per each rewarded action), see Figure 1. We first look at the differences among the five rules that have been used:post-post synaptic rule: after 100 epochs of training, the agents choose the correct response for the first choice in about 77% of trials and exhibit a good performance, with a small standard deviation among agents.post-pre synaptic rule: this rule brings about quite a similar result as the previous one.pre-pre synaptic rule: in spite of the duration (100 epochs), we have poor training. This condition prevents the agents from choosing the correct response for the first choice, with an increase in the number of responses for the second choice. Moreover, variability among agents is large.ex-or covariance rule: the results shown are similar to those obtained with the pre-pre rule.Oja rule: by using the Oja learning rule, the performance is quite similar to that of pre-pre and ex-or rules. However, we notice a smaller variability among agents.

**Figure 1 ijms-23-03452-f001:**
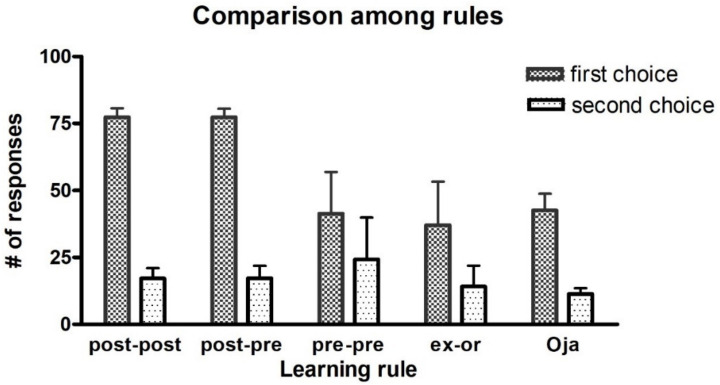
Test trials after 100 epochs of training. Number of correct responses for the first choice (i.e., the choice with probability reward as high as 0.8) and for the second choice (probability of reward as low as 0.3) obtained by performing 100 different test trials after 100 epochs of training, performed with the different training rules. The mean values and standard deviations refer to 10 different independent learning procedures (trained with different noise realizations). It is worth noting that the term “first choice” refers to two different possible actions, depending on the input stimulus used (the action 1 is more frequently rewarded for the stimulus S_1_, and the action 2 for the stimulus S_2_). Each stimulus was used 50 times during the 100 test trials.

For the last three rules, the total number of responses for the first choices is significantly below 100. This signifies that in many trials no cortical neuron won the competition and the network did not select any response.

The entire history of the test carried out after training with the two best rules in different configurations of synapses, is presented in Figure 2.

Starting from a naïve condition, in which each stimulus has about 50% of probability to be chosen, the agents progressively learn to discriminate between the two choices. In particular, if poor synapses are used to perform the test, the agents are unable to distinguish the best choice (at the beginning, the probabilities are close to 50% per each choice). After 100 epochs we can see a quite good performance. After 200 epochs of training, a further performance improvement occurs, displaying an increase in the number of more correct responses (first choice) and a decrease in the number of second choices. The standard deviation remains quite small during the whole history of the test. By using the post-pre rule (right panel), a quite similar outcome is achieved.

On the ground of these results, no significant differences can be observed between post-post and post-pre rules. However, when performing the same test with 4-actions (the results will be presented later), the post-post rule outperforms the post-pre one. For this reason, we choose to present the subsequent simulations with the post-post rule.

To better illustrate the training procedure, the effect of basal training on synapses coming from the sensory cortex (S) to the Go neurons, is given in Figure 3. We have chosen these synapses since they illustrate the relationship between the external context (S) and the chosen activity (Go). The first row of Figure 3 shows the temporal pattern of the synapses entering into the first channel; the second row represents synapses entering into the second channel. The columns represent the corresponding inputs. Since we more frequently rewarded the first action with the stimulus S_1_ = [1 0.3] given as input, and the second action with the stimulus S_2_ = [0.3 1], the network should be able to associate the first element of S with the first action, and the second element of S with the second action. Accordingly, the synapses on the main diagonal strengthen, whereas those out of the main diagonal weaken. This means that the agent can progressively discriminate between the two choices selecting the correct association (stimulus linked to the chosen action). The effect of saturation concerning the synapses on the main diagonal can be seen starting from epoch 170 and epoch 130, respectively. We must point out that synapses can’t overcome a saturation value of 0.8.

As described in the Section 4, after the training, we performed a reversal learning (by inverting the probability of reward for the two actions with each stimulus), starting from different epochs of previous training. In particular, we tested the case of a reversal learning performed after 50 epochs of basal training (moderate learning) and after 75 epochs of basal training (good learning). However, as shown in Figure 4, simulations demonstrated that, in order to reverse the choices, the punishment dopaminergic drop (*D_drop_* in Equation (6), in the Section 4) during reversal learning must be significantly stronger than the value used during the previous basal training. Figure 4 illustrates the results, where the dopamine drop used during the reversal learning (that is, the strength of punishment) is plotted on the x-axis, while the number of correct choices after 200 epochs of reversal learning are plotted on the y-axis.

During basal training, the synapse strengths are modified (see Figure 3), reaching, thus, different performances. In particular, when reversal starts after 50 epochs, the agents have, on average, 65% of correct responses; after 75 epochs they reach an almost successful performance (75% of correct responses), see left panel of Figure 2. As a consequence, starting reversal after a longer training (after 75 instead of 50 epochs), requires an increase in the punishment to forget the previous synapses. In particular, after 50 epochs, the lowest phasic drop in dopamine (*D_drop_* = 0.5) is not enough to perform the task correctly (only 20% of correct responses are achieved after reversal training). If the phasic dopamine drop during punishments is increased (*D_drop_* = 1.0), a good performance is reached (61% of correct responses). With a further increase in dopamine drop (*D_drop_* = 1.5), the agents only slightly improve their performance (66% of correct responses). If the reversal starts after 75 epochs of training, when actions are already well-differentiated, the agents are not able to perform the reversal task properly, neither with the lowest value of *D_drop_* (0.5) nor with *D_drop_* = 1.0. A further increase in punishment is required (*D_drop_* = 1.5) to carry out a more adequate task (35% of correct responses); the performance, however, still remains significantly below that obtained after the basic training. Hence, this preliminary analysis leads us to conclude that a punishment reinforcement is required to improve the reversal training.

In order to understand the previous results, Figure 5 displays the effect of punishment on the activity of the Go neurons in a naïve condition (i.e., before any training) with *D_drop_* = 0.5 (left panel), the effect of a punishment when the reversal starts after 50 epochs with *D_drop_* = 1.0 (central panel), and a punishment when a reversal learning starts after 75 epochs with *D_drop_* = 1.5 (right panel). The red line refers to the winner Go neuron (i.e., the one in the selected action channels) and the blue line to Go neuron in the loser action channel. The dashed black line represents the threshold for the Hebb rule in the post-synaptic neuron, set at 0.5.

In the naïve condition, the activity of the Go neuron in the winner action channel settles at a value close to the threshold (indeed, parameters in the naïve network were given to reach this steady state level); hence, even a moderate phasic punishment can lead the Go activity well below the threshold, leading to a depotentiation of the synapse. Conversely, after training, as a consequence of the increased synapse values, the activity of the winner Go neuron, in an action channel previously reinforced, reaches higher levels. This is evident both in the central and right panels of Figure 5. Hence, a stronger punishment is required to move the level below the threshold, to have a significant synapse depotentiation. In fact, if the phasic dopamine change does not produce evident variations on the Go activity, and if the Go activity is too far from the threshold, the synapses exhibit only a minor depotentiation or can even be paradoxically reinforced. In conclusion, the further the Go activity is from the Hebb rule threshold, the more the punishment should be strong to produce a reduction in the synapse.

A summary of the entire history of the test performed after reversal training is given in Figure 6. It is worth recalling that the number of responses at epoch 1 after 50 and after 75 epochs (Figure 6) corresponds to that of Figure 2 at epoch 50 and at epoch 75, respectively, as that is when the reversal starts.

During reversal, the previous correct response (first choice basal) becomes the more often punished (first choice reversal), whereas the previous more often punished response (second choice basal) is the more often rewarded (second choice reversal). Looking at the left panel (reversal after 50 epochs) we can see that, at first, the number of first choices reversal is higher than the number of second choices reversal. Starting from epoch 125, after training the agents are able to perform a switch between actions, inverting them. Correct responses (second choices reversal) gradually become higher, whereas a reduction in the number of more often punished responses (first choices reversal) is evident. The performance continues to improve until a sort of saturation is reached at epoch 350 after training. It is worth noticing that the standard deviation increases during training, in particular as regards the second choice reversal. This means that some agents find difficulties in reversing the choices, while other agents perform much better than in the mean behavior. After 75 epochs (right panel in Figure 6) the reversal requires a longer time to be achieved. The number of first choices reversal progressively decreases during epochs, and the number of correct actions (second choice reversal) exceeds the number of responses for first choices reversal at epoch 375. Again, the standard deviation increases during training. It is worth noting that, especially if reversal starts after 75 epochs, the final number of responses is well below 100 (in fact, we have 38.7% of correct responses (second choice reversal) and 25.1% of wrong responses (first choice reversal), see the right panel in Figure 6). This signifies that, after a prolonged reversal, the network does not provide any response in about 36% of trials. This is the consequence of an excessive conflict between the old and the new choices.

The pattern of the synapses coming from the sensory cortex (S) to the Go neurons during reversal learning is shown in Figure 7. The reversal learning starts after 50 epochs of training for a total of 400 epochs. In the present case, we use inverted stimuli as inputs. Since the agent now must associate the second stimulus to the first action and the first stimulus to the second action, the synapses out of diagonal increase their strength, reaching the maximum saturation level at around 200 epochs. The synapses on the main diagonal, which were potentiated during the basic training representing the previous first choice, progressively weaken, after an initial plateau phase. Set against Figure 3, we may notice the switch between actions.

### 2.2. Four-Action Choice

In the following section, we describe the results of some simulations performed on 10 independent learning procedures, during a four choices probabilistic task. The whole history of the test performed after training with the two best rules identified for two channels, is presented in Figure 8. The training consists of 300 epochs. Each agent performs 200 trials (50 actions per channel). In the case of the post-post rule (left panel), again starting from a naïve condition represented by the test performed at epoch 1, the agents increase their performance in the simulations run in the subsequent epochs. In fact, we can notice that the number of first choices continues to grow, whereas the number of second choices is reduced.

The right panel shows the results obtained by using the post-pre rule. During the test phase, the performance worsens compared to that obtained with the post-post rule, with reference to first choices. In addition, we can see an increase in the standard deviation values. The number of second choices still remains quite the same.

A possible explanation of the differences between the rules is presented in the Appendix A.

Since with four channels, the post-post rule is the best option, we decide to show the simulations regarding reversal training using that rule. In the following simulations (Figure 9), each panel represents the reversal training performed after 100 (left panel) and 150 (right panel) epochs of training. In this case also, the first reversal was performed with a phasic drop in dopamine as large as 1.0, and the second with a phasic drop as large as 1.5.

Recalling that both in the left and the right panel (Figure 9) the results at epoch 1 are the same as those at epoch 100 and epoch 150 (Figure 8 left panel), respectively, we can now explain the effect of the reversal learning technique with four channels.

Looking at the left panel, the agents succeed in selecting the correct choices, starting from epoch 200 of the reversal phase. The performance is further enhanced as the synapse training progresses, with the mean value of the number of second choices increasingly up to 103/200 with a significant reduction of first choices (36/200). The comparison of the two panels points out that if the reversal starts too late, the agents are not able to perform a proper switch of the choices, even using *D_drop_* = 1.5. Indeed, the number of first choices is slightly decreasing (47.7/200), but the number of second choices does not increase too much (59.1/200).

It is worth noting that, in both panels of Figure 9, the total number of responses is well below 200 (even by considering the residual responses of actions three and four, not shown to avoid cumbersomeness). This signifies that, in many cases, due to excessive conflict, the network does not select any single response during the test.

### 2.3. Sensitivity Analysis on the Tonic Dopaminergic Input

Since dopamine plays a pivotal role in our learning procedure, and several studies show that learning is affected by the dopamine level (for instance learning impairment is evident in patients affected by Parkinson’s disease), in the following we show the effect of a change in the tonic dopamine on training in the two-actions task. Figure 10 displays the responses obtained with the post-post rule during a test trial performed after 100 epochs of training (i.e., the same as in Figure 1) as a function of the tonic dopamine level. However, it is worth noting that, since we assumed that the phasic dopamine peak during reward is twice the basic tonic level, and that the phasic dopamine drop during punishment is half the tonic level, a change in tonic dopamine also affects rewards and punishments.

In the left panel, it is assumed that the tonic dopamine level is decreased during training, but a normal dopamine level is used during the test phase. This represents the case of a Parkinsonian subject, with a depletion in dopamine during training, who performs a test under levodopa medication (thus restoring the normal tonic dopaminergic level). It is evident that, if the tonic dopaminergic input is significantly decreased only during training (from 1.0 to 0.6), the agent responds in almost 100% of trials, but he/she is not able to discriminate between the first and second choices (first choice 54.4/100, second choice 42.5/100 when D = 0.6).

The right panel represents the case when a subject exhibits a small dopamine level both during training and test trials. In this condition, not only is the agent unable to discriminate between the two choices (impaired training), but also exhibits a great number of no-responses during the trials (first choice 24.0/100, second choice 26.2/100 when D = 0.6). The high number of no-responses is due to the low activity level in the Go channels, induced by a depleted dopaminergic input.

### 2.4. Automatic Adjustment of the Phasic Dopaminergic Input

The previous results clearly demonstrate that the more difficult the task is (in particular during reversal learning), the stronger the dopaminergic drop required during punishments to forget a portion of the previous synapse values is. Indeed, if the dopaminergic punishment is not strong enough, both the synapses representing the first choice and the second choice remain too high; causing a conflict between the two choices with many no-response cases (see Figure 2 and the right panel in Figure 9). To avoid this problem, in the previous simulations we manually adjusted the dopaminergic drop, assuming that these changes reflect a top-down control (i.e., a kind of actor-critics instantiation).

Many results in the literature (see [42,43]) suggest that the dopamine phasic response during reinforcement learning not merely reflects the amount of reward and punishment, but above all the expectancy of a reward. Looking at Figure 5, which displays some examples of punishments in Go neurons during different trials (naïve network, reversal after 50 epochs, reversal after 75 epochs), we can suggest an automatic strategy to adjust the phasic dopamine:We can assume that the activity of the Go neuron immediately before the punishment or reward (the level which is shown with an asterisk in Figure 5) represents an *expectancy of the reward* (or the probability of a reward, in fact, in our model the neuron activity is normalized between 0 and 1). If the winner Go neuron at the moment of the reward/punishment is close to 0.5 (which also represents the threshold for our Hebb rule), as in the left panel of Figure 5, the expectancy of reward is about 50%. A high value of the Go activity (as in the middle and left panels in Figure 5) signifies a high expectancy of a reward. This should be associated with a smaller dopamine peak in case of reward, but with a higher dopamine drop in case of punishment. Conversely, a low level of the Go activity in the winner channel represents a low expectancy of a reward, which should be associated with a higher dopamine peak or a smaller dopamine drop. *go_j_*(*t*) represents the activity of the Go neuron in the jth channel (*j* = 1,2 in case of a two-choice task, *j* = 1, 2, 3, 4 in case of a four-choice task). Hence, we can assume a reward expectancy (i.e., the estimated probability of a reward, say $r_{expected}$) as follows:(1)$$r_{expected}=go_{w}\left(t_{response}\right)$$
where *w* represents the winner channel, and $t_{response}$ the instant at which a reward/punishment is given (some examples are shown in Figure 5).In case of a *reward*, the phasic dopamine peak is a non-linear function of the difference between $r_{expected}$ and 0.5. The higher this difference, the smaller the dopamine peak and vice versa. We can write:(2)$$\Delta D_{reward}={\left[2*\left(1-r_{expected}\right)\right]}^{m}\cdot D_{tonic}$$
where *m* represents a parameter, greater than 1, chosen empirically, and *D_tonic_* is the tonic dopaminergic input. The higher *m*, the stronger the effect of an unexpected vs. an expected reward. As it is clear from Equation (2), when the expected reward probability ($r_{expected}$) is 0.5, the phasic dopamine peak is equal to 1, that is the same value we used in all previous simulations. If $r_{expected}$ is close to 1 (in case of a strongly expected reward), the phasic dopamine peak decreases dramatically. If $r_{expected}$ is close to zero, we have a much stronger phasic dopamine peak.In case of a *punishment*, the higher the expected reward, the stronger the phasic dopamine drop should be, to modify the choice. We can write:(3)$$\Delta D_{punishment}=-0.5\cdot {\left[2\times r_{expected}\right]}^{m}\cdot D_{tonic}$$If the *expected reward* is 0.5, the phasic dopamine drop is half the tonic dopaminergic input, i.e., the value used in the previous basal trials. If $r_{expected}$ increases, the phasic dopaminergic drop increases dramatically, thus automatically implementing the major dopamine drop necessary in the reversal tasks.


In the following (Figure 11), we show the results obtained by performing the two-action choices, and four-action choices in the basal and reversal training, by using Equations (1)–(3) to implement an active top-down control of the phasic dopamine changes. The values of *m*, the exponent in Equations (2) and (3) were empirically set at two. We also performed simulations with the value of *m* set at one. If a value *m* = 1 is used, the reversal learning becomes more difficult, as shown in Appendix A.

The results show that the model can perform the basal and reversal learning quite well in case of a two-action task (upper panels). In case of a four-action task, the model can perform the basal task and the reversal task after 100 epochs quite satisfactorily. The reversal task after 150 epochs is still difficult, but significantly improved compared with the one in the right panel of Figure 9.

The complete pattern of the synapses, both during the basal and the reversal phase, is shown in the Appendix A.

### 2.5. Effect of Different Probabilities

As specified in the Section 4, the previous simulations were repeated using different probabilities for the two-action task. In particular, the network was trained and tested using probabilities 0.9 and 0.2 (simpler task) and the probabilities 0.7 and 0.4 (more challenging task). As expected, during the basal training phase, the network learns the correct behavior more rapidly in the simpler task and more slowly in the more complex task. Differences are especially evident after 100 epochs. However, after a longer number of epochs (for instance, 200 epochs of training), these differences are reduced. More significant differences are evident during the reversal learning. In the simpler task (with probability 0.9 and 0.2), reversal learning is more efficient and robust. Conversely, reversal learning is weaker in the difficult task (probabilities 0.7 and 0.4). It is characterized by a large number of no-responses and a more significant standard deviation (hence, large variability among actors).

## 3. Discussion

*General considerations:* in the present work, we simulated probabilistic reinforcement learning and reversal learning in a human BG model, in order to analyze two fundamental problems, not completely solved in previous works:which learning rule is more suitable in a task where the agent must associate different stimuli S with different actions to maximize rewards;which reinforcement signal (i.e., phasic dopaminergic change) is more appropriate to perform a reversal learning procedure, when a subject should forget a part of the previous synapses in favor of new synapses, to avoid a conflict between the previous choice and the new choice.

A third problem, which we only partially analyze in the present paper, concerns the effect that pathological modifications can have on learning capacity. Indeed, many indications emerging from the present findings can be applied to the study of neurological and neuropsychiatric disorders associated with alteration in the dopamine reward system.

It is important to stress that the strategy for training we used in the present study differs from many tasks commonly adopted in probabilistic reversal learning paradigms. In classical forced-choice tasks (see for instance [34,36,44,45]), one stimulus is designated as the correct stimulus, associated with the higher probability of a reward, whereas the other stimulus is associated with a smaller reward probability. The two reward probabilities are then changed during the reversal phase. Hence, on each trial in one phase, the agent must designate only one correct action. Even in studies using a larger number of stimuli (see for instance Peterson et al. [8], who used four different images) subjects should designate only one stimulus, that is the most likely to be rewarded. Our scenario is much more demanding, since at any phase of the experiment, the model should be able to associate a different stimulus with a different action. We think that our paradigm simulates several important situations in real life, in which different actions can be chosen depending on the particular context, and then flexibly modified as the environment changes.

*Hebb rule*: by simulating this task in a probabilistic environment, a first important result emerged, i.e., that the Hebb rule performs in a largely different way, depending on the choice of the post-synaptic or pre-synaptic terms. In particular, our simulations demonstrate that a rule in which the synaptic changes occur only when the post-synaptic Go neuron is active outperforms the other rules.

It is worth mentioning that, in our previous papers devoted to the analysis of bradykinesia in Parkinson disease [40,41] we used a Hebb rule based on the pre-synaptic activity (named *pre-pre* in the present work); this rule worked fairly well in a deterministic scenario, as that used in our previous papers, but fails to work in a probabilistic scenario, as the one simulated in the present study.

It is not easy to understand why a rule performs better than another in a probabilistic task. Some examples are given in the Appendix A, to illustrate the reasons for some of the observed differences. Basically, if a network starts from a naïve condition, (and so at the beginning, the number of errors are quite similar as the number of correct choices, in a two-choice task, or greater than the number of correct choices, in a four-choice task), the use of a pre-synaptic positive part results in a more frequent weakening of useful synapses (i.e., those synapses which instead should be reinforced) compared with the use of a post-synaptic positive part. These differences between the Hebb rules are always evident during the basal learning period in a two-choice task (see Figure 1) and become even more evident if a four-choice task is performed (see Figure 8).

*Dopamine changes and reversal learning*: simulations concerning the reversal learning show a different aspect. Flexibility requires that a subject partially forgets some of the previous synapses, now associated with a wrong choice, in favor of new synapses, pointing to a more rewarded choice. If previous synapses are not dramatically reduced, an excessive conflict emerges between alternative possible actions, with the effect of either a frequent wrong choice or a frequent no-choice. The importance of forgetting some synapses during reinforcement learning has been stressed by Morita, Kato et al. in a series of modeling papers [37,38], although they did not investigate reversal learning specifically. In order to depotentiate previously learned synapses, our simulations point out the necessity for a greater phasic dopaminergic fall during reversal learning, compared with the value used during the initial training; moreover, when the reversal learning becomes more difficult (i.e., in case of a longer previous training and stronger previous habits), a deeper phasic dopaminergic drop is required. This empirical result provides the foundation for the introduction of a top-down control of the dopaminergic input, which should be automatically adjusted to meet the necessity of learning. Actually, as shown in Figure 7, a decrease in previously learned synapses is the more relevant and more complex target of reversal learning, which can jeopardize the overall process if not sufficiently accomplished.

*Top-down dopamine regulation*: the necessity of a top-down control of dopamine phasic changes during reinforcement learning is a subject of active research today, a problem which is not completely solved yet and is strictly related with the flexibility of our behavior and with the management of uncertainty (see [46]). There is large consensus that the striatum plays a pivotal role in this process under the top-down influence of the ventral PFC (especially the OFC) [17,47,48]. Various results suggest a mechanism for an active control of phasic punishment. Ghahremani et al. [14], through a neuroimaging study, observed an increase in the activity of the OFC and of the dorsal anterior cingulate cortex during reversal learning, and inferred that these regions support an inhibition of learned associations during reversal. Albein-Urios et al. [19] found that inhibition of the ventro-lateral PFC, achieved via transcranial direct current stimulation, was associated with higher perseverative errors compared to sham conditions. Cools et al. [5], using functional magnetic resonance imaging, observed signal changes in the right ventrolateral prefrontal cortex on trials when subjects stopped responding to the previously relevant stimulus and shifted responding to the newly relevant stimulus. Recording of single neurons activity in the OFC show that neurons track alterations of reward contingency [49,50]. Several studies demonstrated that lesions of the OFC impair reversal learning [15,16].

All previous considerations underline that reversal learning paradigms require a dynamic reward representation, including an expectation for the possibility of a change. Soltani and Izquierdo [1] recently stressed that update of expected rewards necessitate integration of signals among multiple brain areas and suggested that learning must be scaled up when unexpected uncertainty is high. This dynamic representation, in turn, should be converted into dynamic phasic changes in dopamine [51].

Indeed, many studies suggest that dopamine phasic changes represent the expectancy of a reward, rather than a reward per se (see among the others [42,52,53]) In several previous computational models, this dynamical reward expectancy is simulated using the well-known Sutton and Barto Temporal Difference algorithm [21,54]. Moustafa et al. [36] used this algorithm to implement a critic-actor schema in which the reward prediction is realized through an additional critic node in the network. Conversely, Frank and Claus [20] adopted a working memory layer to simulate the OFC. A working memory m odule is also used in the model by Deco and Rolls [55].

*The new formulation for phasic dopamine changes*: our preliminary simulations clearly demonstrate the need for an ad hoc adjustment in the phasic dopamine signal, in order to adapt this signal to the difficulty of the learning process. Accordingly, in the last part of this work, we propose a new strategy to compute the *expected reward*, which is not shared by other models (at least to our knowledge) and does not require any additional computational node. Our suggestion is that information on the *expected reward* can be obtained directly from the activity of the winner Go neuron, immediately before the reward/punishment is given (see Equation (1) and Figure 5). At odd with other models, our rule provides a context dependent reward expectancy. This information can then be directly manipulated by a top-down control, implementing a critic-actor schema, which computes the phasic dopamine changes (i.e., implementing Equations (2) and (3)) in a dynamical way. Of course, our Equations (2) and (3) are empirical, since they simply compute the m-power of the difference between the predicted reward and a naïve prediction (set at 0.5). An important role in our model is given by the parameter *m*.

There are two possibilities, which can be tested in future work: either that a single value of *m* provides adequate reversal learning both in simple and challenging conditions; or that the value of *m* must be automatically adjusted by a critic. A value *m* = 2 seems appropriate in a two-action task with probabilities 0.8 and 0.3 and also in the case of simpler tasks (unpublished simulations). An increase in *m* may be required if the task becomes excessively challenging. This will be the subject of future study.

*Comparison with existing data*: since, as specified above, our task is quite different from the reversal learning tasks found in the literature (which involve one forced choice), we did not make a direct comparison with existing data. However, some considerations can be done, which may represent future testable predictions. Firstly, the reversal learning phase generally requires more time compared with the analogous initial learning phase (see for instance Figure 11). Moreover, the number of perseverative errors remains higher during the initial period of the reversal. These results are qualitatively in accordance with those of previous experimental tests [8,15] and of previous computational results [36,39] although, in those works tests were performed in simpler conditions.

All of these results are quite understandable since, during reversal learning, the agent must forget previous synapses and simultaneously reinforce the new ones. Indeed, looking at the synapse patterns in Figure 7, one can see that about 150–200 epochs are required before a significant decrease in the previously reinforced synapses.

Another important result that deserves attention is that, after a prolonged reversal learning, the agent sometimes fails to provide any response, especially if reversal occurs in the four-choice task after a long previous learning phase. That is due to an excessive conflict between the previous and the new choice.

*Pathological conditions*: a notable future application of our model concerns the simulation of learning impairment in neuropsychiatric and neurological diseases. Mukherjee et al. [56] showed that depressed individuals make fewer optimal choices and adjust more slowly to reversals. Schizophrenic patients perform similarly to control individuals on the initial acquisition of a probabilistic task, but show substantial learning impairments when probabilities are reversed, achieving significantly fewer reversals [17]. Probabilistic reversal learning is impaired in youths with bipolar disorders compared to healthy control subjects [57]. Bellebaum et al. [6] observed that patients with selective BG lesions need more time than control subjects to perform a reversal task. Several of the previous results can be qualitatively simulated with our model by reducing the parameter *m* (or more generally by reducing the phasic dopamine drop); this possibility opens interesting perspectives towards future model applications in clinical scenarios.

While the previous deficits seem especially concerned with a malfunctioning of the ventral PFC, learning deficits in patients with Parkinson’s disease (PD) are likely to be connected with a reduction of the baseline dopamine level. Accordingly, reinforcement learning is impaired in PD subjects [8,58]. However, dopaminergic therapy improves some cognitive functions and worsens others in patients with PD, an apparently paradoxical result [9]. In particular, several authors observed that medication worsens reversal learning in PD subjects [7], although this effect seems to depend on the disease severity [9,44].

In the present study, we simulated the effect of a change in dopamine basal level (Figure 10). Our model suggests two concomitant phenomena at the basis of a learning failure in PD subjects, the first related with learning (i.e., a poor adjustment of synaptic changes) and the other related with the strength of the mechanism during the test phase (related to the tonic dopamine level during the test). If the dopamine level is reduced only during the task, but is restored during the test trial, the subject exhibits impaired learning, but with almost 100% of responses. Conversely, if dopamine levels are low both during the training and the test phase, two concurrent phenomena become evident: the incapacity to discriminate between the two choices and a frequent occurrence of no-responses (which corresponds to a typical deficit of PD subjects to start actions, producing symptoms like freezing or bradykinesia). This represents a testable prediction of the model, that may be investigated in future studies, and is in line with the ‘integrative selective gain’ framework proposed by Beeler ([59,60], but see also [41,61]).

*Model limitations*: in the present work, we assumed that the unique effect of the PFC top-down control consists in dynamic phasic dopamine changes (especially during punishments). This is the only input to the BG module (besides the external stimulus S). It is possible that other parameters of the model, not only D, are under active control from the PFC (such as the learning rate or the threshold in the Hebb rule). Indeed, Metha et al. [62] suggested the use of different learning rates for negative and positive prediction errors. Harris et al. [63] assumed a higher learning rate during reversal learning. An adjustable threshold in the Hebb rule, depending on neural average activity, is typical of the Bienenstock, Cooper, and Munro learning rule, often used in neural network trainings [64]. However, our model may be seen as more parsimonious, assuming that the entire learning variability depends on the dopaminergic input only. It may be of value to change the learning rate and/or threshold in future works, if the comparison with experimental results should require an improved learning capacity.

Furthermore, the Go pathway plays the more relevant role in our model, while the NoGo synapses increase monotonically (although with a different rate for the different choices, see Appendix A), signaling a large number of punishments. The choice of a different learning rule for the NoGo portions of the striatum may further improve learning, achieving a better balance between Go and NoGo.

Finally, in future work the model should be tested against human behaviors in the same tasks to compare changes in human performance across training epochs with those predicted by the proposed model. Simulation of experimental results, however, is likely to require an ad hoc fitting of some parameters (for instance the value of the learning rate and the parameter *m*) to adapt the general network to the particular behavior of individual subjects.

In conclusion, we wish to stress that the present model does not only provide a new step towards a better understanding of dopamine-based reinforcement learning in the PFC-BG complex, but can also represent a tool for future Artificial Intelligence algorithms. This means that it can be applied, in a flexible way, to all conditions where actions must be associated with stimuli in a non-stationary environment.

## 4. Methods

### 4.1. Structure of the Network

The overall network is inspired by the human BG. Each neural unit in the model is described using a first-order differential equation, which simulates the integrative properties of the membrane, along with a sigmoidal relationship, which reproduces the presence of lower threshold and upper saturation of neuron activity.

The specific structure of the network is depicted in Figure 12, where two channels are implemented. A similar structure can be used with four channels, to simulate a four-action choice. The model includes a sensory representation (S) and the corresponding motor representation in the cortex (C). The latter considers several actions in mutual competition, each represented by a segregated channel. The model can be used to simulate several tasks involving reinforcement learning, in which the agent learns to perform a task under the pressure of an environment which rewards or punishes.

In the present study, we focus on probabilistic learning, in which a given action is not always rewarded or punished, but rewards and punishments are delivered per each action with a given probability. Moreover, as specified in the introduction, all actions can be chosen depending on the particular stimulus from the sensory cortex and much focus is given on reversal learning, to study how the network can flexibly change its behavior after a change in the given probabilities.

Downstream the cortex, the model includes the striatum, functionally divided according to dopamine receptor expression (D1: Go, or G, D2: NoGo, or N), the subthalamic nucleus (STN), the globus pallidus pars externa (GPe, or E), the globus pallidus pars interna (GPi, or I), and the thalamus (Th). Accordingly, the model includes the three main neurotransmission pathways (direct, indirect, and hyperdirect) used by the BG. In absence of sufficient stimuli, all actions are inhibited, since the GPi receives a strong basal input and inhibits the thalamus. More particularly, in basal conditions, the cortex, the thalamus, and the striatum are inhibited; conversely, the GPi and GPe exhibit a certain basal activity. We assumed that the basal activity of the GPe is at about half the maximal activity; conversely, the basal activity of the GPi is higher, close to the upper saturation, in agreement with physiological data [65]. This activity is necessary to maintain the thalamus inhibited, until a significant disinhibition occurs in the GPi as a consequence of the activation of one Go pathway [40].

In the presence of a sufficient external stimulus, the cortex can select a response based on a competition among cortical neurons. This competition is realized with a winner-takes-all (WTA) process, implemented via lateral inhibition in the cortex. However, in order to win the competition, a neuron in a WTA circuit must receive a strong self-excitation: this is an essential mechanism to trigger the instability that is typical of WTA dynamics. In the model, this self-excitation becomes operative only when the corresponding neuron in the thalamus is activated: this thalamic neuron sends excitation to the cortex and closes a self-excitatory loop (see Figure 12). Selection of an active neuron in the thalamus, in turn, is a consequence of a prevalence of the Go pathway on the NoGo pathway for a specific action channel. Hence, the final decision actually occurs in the BG, and this choice becomes operative when the corresponding neuron in the thalamus overcomes a certain excitation threshold.

In the model, the hyperdirect pathway (from the cortex to the STN and then to the GPi) is used to excite the GPi (and so to inhibit the thalamus) in the presence of several conflicting actions in the cortex. This is used to prevent many simultaneous cortical winners, and let the cortex have more time to solve the conflict.

In the following we will use the symbol *W^lk^* to represent the array of all synapses entering into a downstream layer, *l*, from an upstream layer, *k*. Hence, the lower case symbol, $w_{ij}^{lk}$, is used to represent an individual synapse in the previous array, namely the synapse from neuron at position *j* in the upstream layer *k*, to a neuron at position *i* in the downstream layer *l*. These arrays have dimension *N_c_* × *N_c_*, where *N_c_* represents the number of channels used in the simulation, equal to the number of possible selected actions (2 or 4). It is worth noting that all these arrays are diagonal, since all channels are segregated (hence, $w_{ij}^{lk}=0\;\mathrm{if}\;i\neq j$), with the only exception of the arrays originating from the sensory cortex (i.e., the array *W^GS^* from the sensory cortex to the Go neurons, and the array *W^NS^* from the sensory cortex to the NoGo neurons), which are fully connected.

### 4.2. The Effect of Dopamine on the Go and NoGo Pathways

According to the literature, we assumed that dopamine exerts a different effect on neurons in the Go pathway (which express D1 receptors) and neurons in the NoGo pathway (D2 receptor expression). In particular, dopamine has an excitatory influence on active neurons in the Go pathway, and an inhibitory effect on the inactive neurons, thus realizing a kind of contrast enhancement mechanism [66], which favors the selection of the winner action only. Conversely, dopamine has an inhibitory effect on all neurons in the NoGo pathway. Furthermore, we assumed that the action of dopamine is potentiated by cholinergic influences, through a kind of push-pull mechanism: an increase in dopamine reduces the cholinergic activity, thus introducing a disinhibition on the Go pathway and further inhibition in the NoGo. In fact, cholinergic activity is inhibitory on the Go and excitatory on the NoGo, while cholinergic changes are always specular to dopamine changes [67].

The effects of dopamine changes in the model are represented through a quantity, named *dopaminergic input*, *D*. The latter exhibits a constant (tonic) level ($D_{tonic}$) at rest, a transient phasic ($\Delta D_{phasic\;}$) increase ($\Delta D_{phasic\;}=\Delta D_{reward}$) during rewards and a transient phasic decrease ($\Delta D_{phasic\;}=\Delta D_{punishment}$) during punishments. As expressed in the following Equations (4)–(6):(4)$$D=D_{tonic}+\Delta D_{phasic}$$
(5)$$\Delta D_{reward}=D_{tonic}$$
(6)$$\Delta D_{punishment}=-D_{drop}\;\cdot D_{tonic}$$

It is worth noting that, in some of the subsequent simulations, the value of the dopaminergic input can become negative during a phasic dopaminergic drop. This is not a contradiction since, in the model, the quantity *D* does not represent dopamine concentration per se, but rather an external input, which accounts for the effect that dopamine concentration changes can have on the neurons in the striatum, mediated via D1 and D2 receptors.

According to Equation (5), following a reward, the dopaminergic input is increased to a value twofold the basal one ($D_{tonic}$) and maintained for a time sufficient for the network to reach a new steady state condition, characterized by a high value of the winner Go neuron and a low value for the other Go and NoGo neurons. Similarly, after a punishment, the dopaminergic input is decreased by a factor *D_drop_*, causing a fall in the activity of the winner Go neuron and an increase in the activity of the NoGo neurons (especially in the winning channel). Finally, in the steady state condition after a reward or a punishment, the Hebb rule described below is applied.

In the last simulations, as described in the Section 2, Equations (5) and (6) are replaced with new expressions (see Equations (2) and (3)), to account for *expected reward* changes.

### 4.3. Synapse Learning

We assumed that synapses entering into the striatum (i.e., from the sensory cortex into the Go and NoGo layers, named *W^GS^* and *W^NS^*, and from the motor cortex to the Go and NoGo layers, *W^GC^* and *W^NC^*, respectively) are plastic and can be trained using Hebbian mechanisms. In order to implement these mechanisms, we assumed that the activity of the post-synaptic neuron (say $y_{i}^{A}$ to denote the activity of neuron *i* in the layer *A*) and the activity of the presynaptic neuron (say $y_{j}^{B}$ to denote the activity of neuron *j* in the layer *B*) are compared with a threshold and multiplied. In this way, both long term potentiation can be simulated, when activity of both neurons is above the threshold, and long term depression, when one neuron activity is above the threshold and the other is below the threshold. Furthermore, the synaptic change is applied only after a reward or a punishment, when a dopamine change occurs.

However, in order to avoid that, a synapse is reinforced when both the pre-synaptic and post-synaptic activities are below the threshold (i.e., when both neurons are scarcely active, a condition which should not produce any synaptic change), we introduced a function “positive part” either to the post-synaptic or pre-synaptic term in the Hebb rule (see Equations (7)–(9)): this signifies that, the Hebb rule should be applied only when the corresponding neuron is active. In this way, different alternative forms of the Hebb rule can be simulated and tested. Asdemonstrated in the Section 2, these different rules have different behavior in the hetero-associative paradigm used in this work.

In previous papers, we used the following version of the Hebb rule, that is similar to a rule used in [40], assuming that the activity of the presynaptic neuron must be above the threshold to have a synaptic change. We can write:(7)$$\Delta w_{ij}^{AB}=\sigma {\left(y_{j}^{B}-\vartheta^{PRE}\right)}^{+}\left(y_{i}^{A}-\vartheta^{POST}\right)$$
where $\Delta w_{ij}^{AB}$ represents the variation of the synapse between the pre-synaptic neuron j in layer B (B = S or C) and the post-synaptic neuron i in layer A (A = G or N) and the ()^+^ represents the function “positive part”, that is: (*x*)^+^ = *x* if *x* > 0; otherwise (*x*)^+^ = 0.

Since the function “positive part” is applied to the presynaptic neurons (both in the Sensory cortex S and in the motor Cortex, C), this rule (Equation (7)) will be named the pre-pre synaptic rule.

However, as shown in the Section 2, and further illustrated in the Appendix A, we observed that the previous rule, when applied to a probabilistic hetero-associative problem, does not work properly. Hence, we tested different rules.

In a rule named post-post synaptic rule (Equation (8)), we assumed that the function “positive part” applies to the post-synaptic neurons, both in Go and NoGo pathways:(8)$$\Delta w_{ij}^{AB}=\sigma \left(y_{j}^{B}-\vartheta^{PRE}\right){\left(y_{i}^{A}-\vartheta^{POST}\right)}^{+}$$

Hence, the synapse change is applied only if the post synaptic neuron is active.

As analyzed in the Appendix A of this work, the previous equations may have advantages and disadvantages when applied to the schema in Figure 12. So, we considered also the possibility that the postsynaptic rule applies to synapses connecting the sensory cortex to the striatum, while the pre-synaptic rule applies to synapses connecting the motor cortex to the striatum. Hence, two alternative rules have been applied depending on the upstream region:(9)$$\Delta w_{ij}^{AC}=\sigma {\left(y_{j}^{C}-\vartheta^{PRE}\right)}^{+}\left(y_{i}^{A}-\vartheta^{POST}\right)\;(\mathrm{from}\;\mathrm{the}\;\mathrm{motor}\;\mathrm{cortex})\;\Delta w_{ij}^{AS}=\sigma \left(y_{j}^{S}-\vartheta^{PRE}\right){\left(y_{i}^{A}-\vartheta^{POST}\right)}^{+}\;(\mathrm{from}\;\mathrm{the}\;\mathrm{sensory}\;\mathrm{cortex})$$
where the post-synaptic neuron i belongs to layer A (A = G or N). In the following, Equations (9) are named the post-pre synaptic rule.

Finally, we also tested two additional more traditional rules. In the first, we applied the covariance Hebb rule, which only excludes the possibility of potentiation when both neurons are below threshold (as in the exclusive or operation). This rule, named the ex-or covariance rule (Equation (10)):(10)$$\Delta w_{ij}^{AB}=\begin{array}{c} \sigma \left(y_{j}^{B}-\vartheta^{PRE}\right)\left(y_{i}^{A}-\vartheta^{POST}\right)\;\;\;if\;\;y_{j}^{B}\geq \vartheta^{PRE}\;\;or\;\;\;y_{i}^{A}\geq \vartheta^{POST}\;\;\;\\ 0\;\;\;\;\;\;\;\;\;\;\;\;\;\;\;\;\;\;\;\;\;\;\;if\;\;y_{j}^{B}<\vartheta^{PRE}\;\;and\;\;\;y_{i}^{A}<\vartheta^{POST} \end{array}$$

Finally, we also tested the Oja rule (Equation (11)), which is a classic Hebb rule (without threshold) with the inclusion of a forgetting factor to have depotentiation:(11)$$\Delta w_{ij}^{AB}=\sigma \left(y_{j}^{B}y_{i}^{A}-y_{i}^{A}w_{ij}^{AB}\right)$$

Finally, for any of the previous rules, we assumed that synapse cannot decrease below zero nor can increase above a maximum value (say, *w_max_*). That is, after any learning step, we checked:(12)$$w_{ij}^{AB}=w_{max}\;\;\;if\;\;w_{ij}^{AB}>w_{max}\;w_{ij}^{AB}=0\;\;\;if\;\;w_{ij}^{AB}<0$$

### 4.4. Training the Network

The network was trained using a probabilistic reinforcement learning paradigm, in which a combination of inputs is given to the network in the sensory cortex and the chosen action is rewarded or punished with a pre-assigned probability. If the stimulus changes, the chosen action also changes. This is a more difficult protocol compared with a case in which an action is always rewarded or always punished following a given stimulus-response combination. It is also a more realistic paradigm, compared with the case when only one forced choice is implemented (i.e., a choice which does not depend on the context or stimulus *S*). Moreover, after a learning period, we modified the probability associated with rewards and punishments, to simulate a reversal learning paradigm; hence, the preliminary choices are necessarily modified on the basis of a new experience in a non-stationary environment.

Since the BG are especially involved in a slow learning of positive and negative outcomes, resulting in motor habits [20,68], we used a slow value for the learning factor (*σ* in Equations (7)–(11)); as a consequence, the training required different epochs to reach a clear discrimination between responses.

It is worth noting that, while the number of elements in the motor cortex always corresponds to the number of possible actions, the dimension of the sensory representation S can be independent of the number of selected actions and can be chosen to simulate inputs with different complexity. In this work, we used a vector *S* with dimension 2 × 1 and 4 × 1, for 2 and 4 channels, respectively, just for the sake of simplicity, to facilitate the analysis of synapses.

#### 4.4.1. Two-Choice Experiment

In a first paradigm, we simulated a two-choice experiment, where one of two actions (described through channels 1 and channels 2 in the model) in response to two alternative stimuli must be selected. A schematic representation of the experiment is shown in Figure 13. The network starts from a naïve condition, in which all synapses have the same values for each channel (hence no action is preferentially selected) and the synapses from each element of the input stimulus are identical (hence the stimuli in the vector S have initially an identical meaning). The values of these initial synapses have been given so that the activity of the Go and the NoGo neurons in the winner channel, before any reward or punishment, settle at a value as high as 0.5 (which is the middle since the activity is scaled between 0 and 1). Rewards increase the activation of the winner Go neuron well above 0.5, while punishments decrease activation to a small value. The opposite occurs for the NoGo neuron in the winner channel. A Gaussian white noise was added to all neurons in the motor cortex (zero mean value, standard deviation 0.2) to have a different response at each trial.

Each training procedure consisted of 200 epochs. Each epoch consists of the two stimuli, S_1_ = [1 0.3] and S_2_ = [0.3 1], permuted in a random fashion. A total of 10 independent learning procedures (each with 200 epochs starting from the naïve network and different noise realizations, with different seed of noise) were performed, to simulate 10 different networks independently trained.

At each trial, a reward or punishment is given whenever the network performs a single choice. This occurs if one neuron in the motor cortex reaches a value higher than 0.9 in the final steady state condition, while all other neurons in the motor cortex have activity close to zero (thanks to the winner takes all dynamics). The cases in which no neuron reaches the value 0.9 (absence of a response), or more than one neuron overcomes the value 0.9 (multiple responses) are considered as no-response, and neither a reward nor a punishment is assigned.

During the basic learning phase, in response to the first stimulus, S_1_, the first action is randomly rewarded with a probability as high as 0.8 (otherwise the action is punished), while the second action is randomly rewarded with a probability as low as 0.3 (otherwise punished). When a second stimulus S_2_ is given the reward probabilities are 0.3 for the first choice and 0.8 for the second one. We also tested different values for the probabilities to implement a simpler task (probabilities 0.9 and 0.2) or a more challenging task (probabilities 0.7 and 0.4). As seen in the Section 2, we presented the results for the intermediate task only, which represents a sufficient level of complexity. A few details on the other tasks have been given at the end of the Section 2.

Reversal learning can theoretically be initiated after any epoch during the basal learning. However, the longer the previous learning period (i.e., the greater the number of epochs used for learning), the more difficult is the reversal. For this reason, reversal learning was attempted at different moments, namely after 50 epochs and after 75 epochs of previous learning. Reversal consists in the same inputs S_1_ and S_2_, randomly permuted at each epoch, but with the opposite associated reward probabilities: action 1 is associated with 0.3 reward probability after presentation of the stimulus S_1_ and with 0.8 reward probability after S_2_; action 2 is associated with 0.8 and 0.3 reward probabilities after S_1_ and S_2_, respectively.

The capacity of the network to learn the more probable reward, still maintaining the capacity to explore different possibilities, is tested via a testing procedure. In total, 100 trials are executed, by using the inputs S_1_ and S_2_ 50 times each, but without rewards or punishments, and the number of responses (either the reinforced action or the unreinforced action) is counted. Correct learning is characterized by a greater percentage number of the reinforced responses, and a smaller number of unreinforced ones. This testing procedure has been performed starting from the network obtained after different epochs of learning (1, 25, 50, 75, 100, 150, 200) to assess the effect of a longer learning on the network behavior. During the test, random Gaussian noise was applied not only to cortical neurons, but also to the input stimuli (simulating a noisy environment).

#### 4.4.2. Four-Choice Experiment

The simulations were then repeated using a more complex 4-actions paradigm. In this case we used the stimuli indicated below, associated with the reward probabilities for the four actions
S_1_ = [1.0 0.3 0.1 0.1] associated with the reward probabilities: P_1_ = [ 0.8 0.3 0.1 0.1]S_2_ = [0.3 1.0 0.1 0.1] associated with the reward probabilities P_2_ = [ 0.3 0.8 0.1 0.1]S_3_ = [0.1 0.1 1.0 0.3] associated with the reward probabilities: P_3_ = [ 0.1 0.1 0.8 0.3]S_4_ = [0.1 0.1 0.3 0.1] associated with the reward probabilities: P_4_ = [ 0.1 0.1 0.3 0.8]

During reversal learning, the probabilities P_1_ and P_2_ were inverted for the stimuli S_1_ and S_2_, and the probabilities P_3_ and P_4_ were inverted for the stimuli S_3_ and S_4_. Each epoch consists of each four possible inputs, given in a randomly permuted order. The test phase consisted of 200 trials, and each input is used 50 times.

### 4.5. Sensitivity Analysis

In order to better understand the role of some parameters in the model, a sensitivity analysis was performed. Indeed, the most important parameters in the model are the tonic dopaminergic input, *D_tonic_*, and the dopamine fall during punishments, *D_drop_* (we remind that the dopaminergic increase after a reward ($\Delta D_{reward}$) was always equal to +*D_tonic_*, i.e., a twofold increase in the dopaminergic input). The parameter *D_drop_* was essential to control learning, especially in the reversal phase, hence its role was tested carefully by repeating learning with different values of *D_drop_*. We assume that this parameter must be under control of top-down regulating mechanisms (see Section 3).

Different values of *D* (tonic condition) were tested, to analyze how learning can be affected by dopamine imbalances, as those occurring in Parkinsonian subjects, or after medication.

### 4.6. Active Control of the Phasic Dopamine

The simulation results suggest that a different phasic dopamine should be used during reversal learning compared with the basic learning, to avoid excessive competition between a previously learned choice and a new choice (in particular unexpected punishments should have a strong impact during a reverse choice, to depotentiate the synapses). Hence, in the last series of simulations shown in the Section 2, we repeated the basic learning procedure and the reversal learning procedure assuming that the phasic dopamine changes are *controlled by the activity of the winner Go neuron*. Briefly, we can consider that the value of the winner Go neuron, at the instant of the reward/punishment, *signals the expected probability of a reward*. This quantity is automatically updated by the network, on the basis of a previous history of rewards and punishments, and also depends on the input stimulus (hence on the context). Consequently, we decided that the higher the Go activity, the smaller the phasic dopamine reward (in case the action is rewarded) or the higher the phasic dopamine punishment (in case the action is punished). 

## Figures and Tables

**Figure 2 ijms-23-03452-f002:**
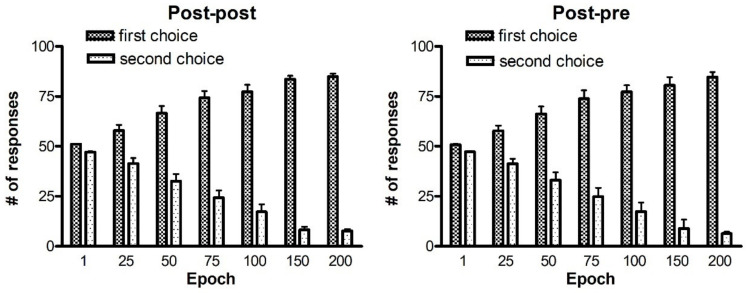
This figure shows the capacity to perform the task correctly (i.e., to select the more correct response, that is the first choice), as a function of the duration of the previous training. The meaning is the same as in Figure 1. (the bars represent mean values and standard deviation on 10 independent learning procedures), but the 100 test trials were performed using the synapse values obtained after different epochs of training. The left panel refers to the post-post rule, the right panel to the post-pre rule.

**Figure 3 ijms-23-03452-f003:**
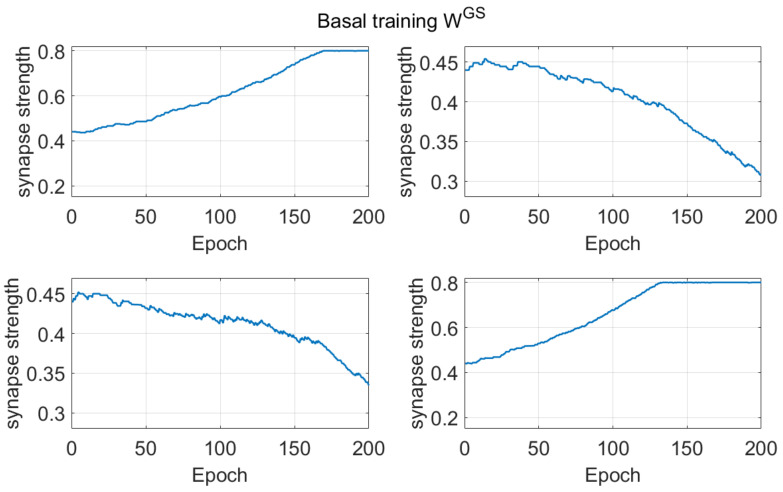
Effect of training on the sensory synapses. The figure shows the temporal pattern of the synapses, which connect the external stimulus S to the Go neurons, as a function of the training epoch (rule post-post). Since we used two action channels, and a 2 × 1 stimulus vector, the synapse array has the dimensions 2 × 2. The first line represents the two synapses entering into the Go neuron in the first channel from the two input stimuli. The second row represents the two synapses entering into the Go neuron of the second channel, from the same input stimuli. By increasing synapses in the principal diagonal, and decreasing those in the secondary diagonal, the network recognizes that a high value of the first stimulus is especially associated with a first-action reward, whereas a high value of the second stimulus more often rewards the second action.

**Figure 4 ijms-23-03452-f004:**
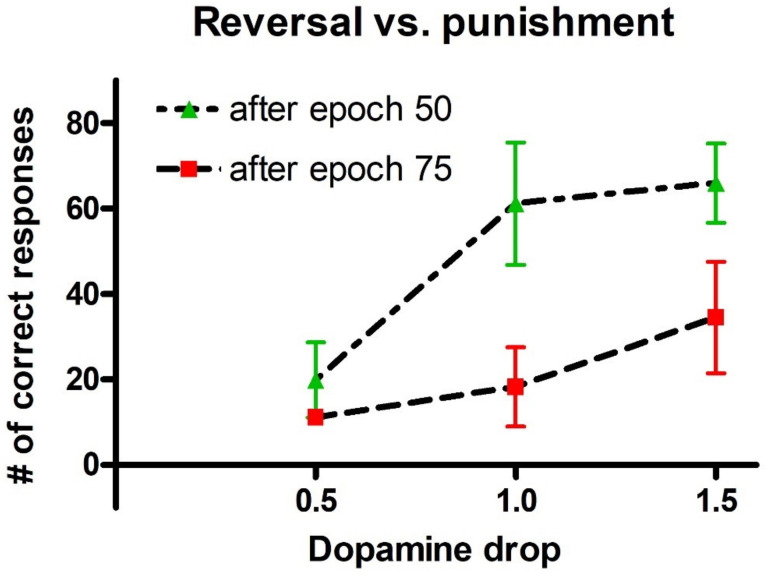
Effect of the dopamine phasic drop used during the reversal learning. The figure illustrates the number of correct responses achieved during the test trials after 200 epochs of reversal learning, as a function of the dopamine drop (i.e., the strength of punishment) used in the reversal training. Two cases are illustrated, when reversal learning starts after 50 epochs of previous basal training (a moderate previous learning) and after 75 epochs (a good previous learning). A dopamine drop as high as 1.0 allows a good reversal learning after 50 previous epochs. However, the reversal learning remains difficult after 75 previous training epochs, even if the dopamine drop is as high as 1.5.

**Figure 5 ijms-23-03452-f005:**
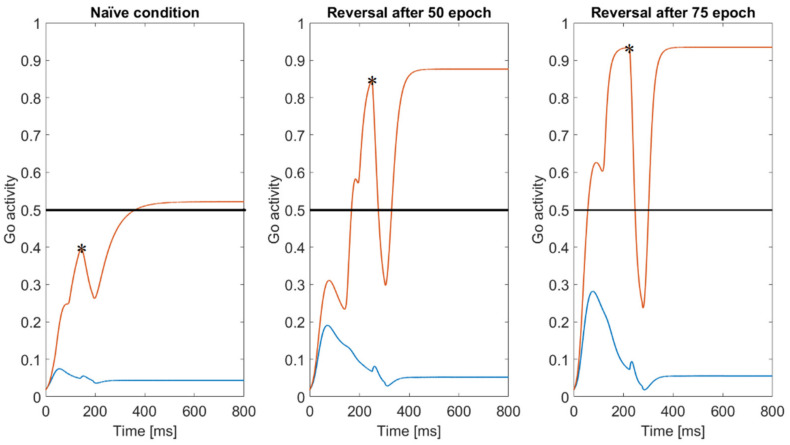
Temporal pattern of the activities of the two Go neurons, in response to a punishment. The red line represents the winner Go neuron (i.e., the one belonging to the selected action) while the blue line represents the activity of the Go neuron in the losing channel. The black horizontal thick line is the threshold for the post-synaptic neuron in the Hebb rule (equal to 0.5). The left panel represents the response in the naïve network (before any training) when the activity of the winner Go neuron settles close to the threshold. Hence, a small punishment is sufficient to achieve a synapse depotentiation. The other panels refer to the network at the beginning of the reversal phase, after 50 epochs of previous training (middle) and after 75 epochs of previous training (right), when an action previously reinforced is now punished. The winner Go neuron, as a consequence of previous learning, settles at a higher value: hence, a stronger punishment is required to move the activity well below the threshold, to allow the depotentiation of a previously learned habit. Finally, the asterisks represent the activity of the winner Go neuron immediately before the punishment, used to compute a value for the expected reward (see Equation (1) below).

**Figure 6 ijms-23-03452-f006:**
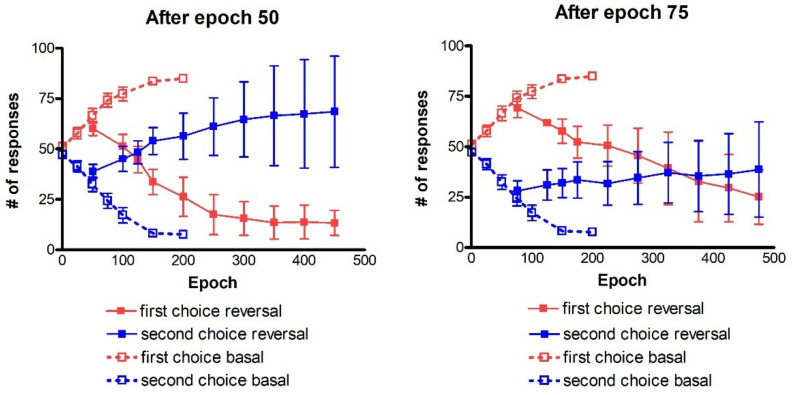
Temporal pattern of the responses in the test trials when the reversal learning is performed after 50 epochs of previous basal training (left panel) and after 75 epochs (right panel). All results refer to the post-post rule. The dashed dotted lines with open squares represent the results of the previous basal training (the same results as in the left panel of Figure 2). The continuous lines with filled squares represent the results of the reversal learning phase. All points are the results of 100 test trials, performed by using the synapses obtained at the given instant. The phasic dopaminergic drop was 0.5 in the basal training, 1.0 in the reversal training performed after 50 epochs, and 1.5 in the reversal training performed after 100 epochs. It is worth noting that the term “first choice” refers to the actions more often rewarded during the basal training with a given stimulus, which becomes more often punished during the reversal.

**Figure 7 ijms-23-03452-f007:**
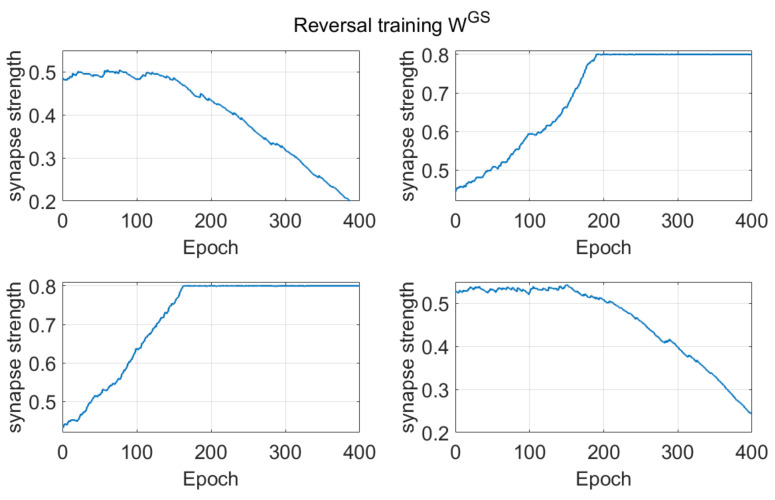
Effect of reversal learning on the sensory synapses. The figure shows the temporal pattern of the synapses which connect the external stimulus S to the Go neurons, as a function of the reversal training epoch (rule post-post) performed after a previous 50 epochs basic training. The meaning is the same as in Figure 3, but now the first channel responds to the second stimulus and the second channel to the first one. Hence, the synapse strength increases in the secondary diagonal of the synapse array, and decreases on the principal diagonal, by inverting the initial levels.

**Figure 8 ijms-23-03452-f008:**
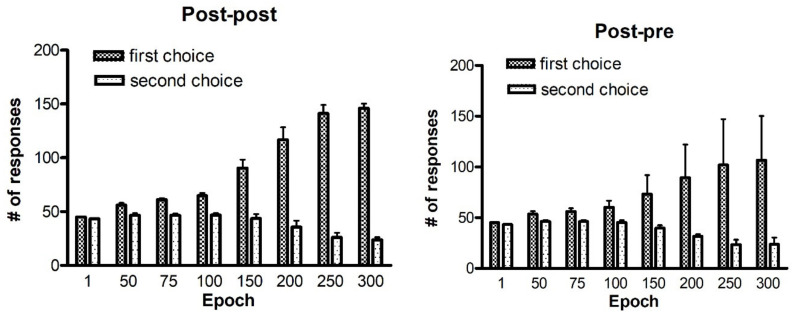
This figure shows the capacity to perform the task correctly in the case of a four-action task (i.e., to select the more correct response, that is the first choice, and the second more correct response, second choice) as a function of the duration of the previous training. Each test consists of 300 trials, in which the stimulus associated with each action is given 50 times. The bars represent mean values and standard deviation on 10 independent learning procedures. The left panel refers to the post-post rule, the right panel to the post-pre rule.

**Figure 9 ijms-23-03452-f009:**
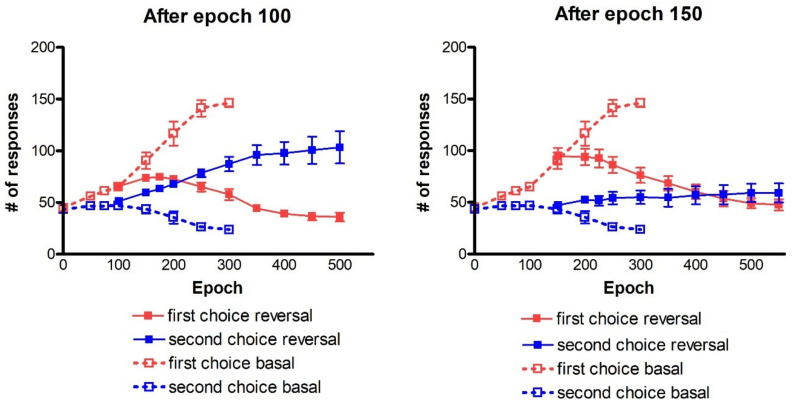
Temporal pattern of the responses in the test trials with reference to a four-action choice, when the reversal learning is performed after 100 epochs of previous basal training (left panel) and after 150 epochs (right panel). All results refer to the post-post rule. The dashed dotted lines with open squares represent the results of the previous basal training (the same results as in the left panel of Figure 8). The continuous lines with filled squares represent results in the reversal learning phase. All points are the result of 200 test trials, performed by using the synapses obtained at the given instant. The phasic dopaminergic drop was 0.5 in the basal training, 1.0 in the reversal training performed after 100 epochs, and 1.5 in the reversal training performed after 150 epochs It is worth noting that the term “first choice” refers to the actions more often rewarded during the basal training with a given stimulus (probability of reward 0.8), which become the second choice during the reversal (probability of reward 0.3). The second choice refers to actions with reward probability as low as 0.3 during training, and 0.8 during reversal learning.

**Figure 10 ijms-23-03452-f010:**
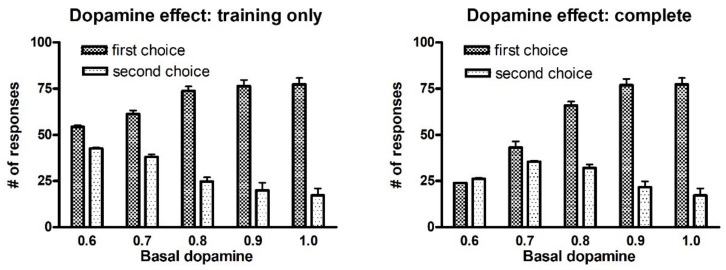
The capacity to perform the task correctly (i.e., to select the more correct response, that is the first choice), in a two-action task as a function of the basic dopaminergic input (post-post rule). Each test consists of 100 trials, in which the stimulus associated with each action is given 50 times. The bars represent mean values and standard deviation on 10 independent learning procedures. The left panel displays the case in which a reduced dopaminergic input is used during training, but a normal dopaminergic input is restored during the test (medication). In the right panel, a reduced dopaminergic input was used both during training and testing.

**Figure 11 ijms-23-03452-f011:**
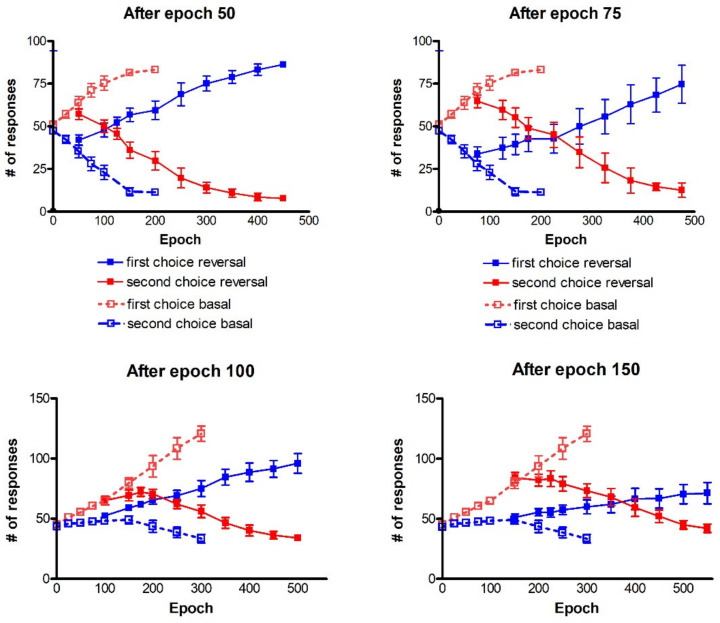
Results of the tests performed after basal training and a reversal learning, obtained by using the Equations (1)–(3) to automatically compute the phasic changes in the dopaminergic input as a function of the expected reward. The panels in the upper row refer to a two-choice task (100 trials per each test) and should be compared with the curves in Figure 6. The panels in the bottom row refer to a four-action task (200 trials per each test) and should be compared with the curves in Figure 9.

**Figure 12 ijms-23-03452-f012:**
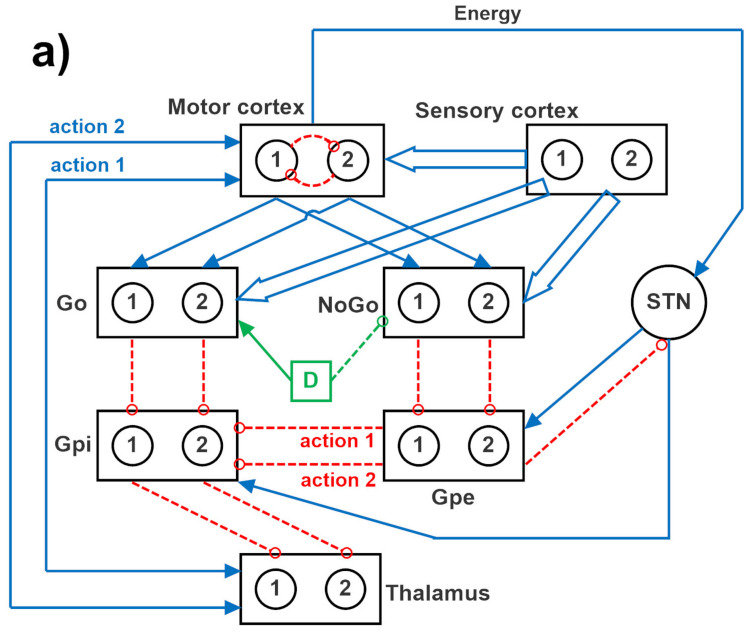
(**a**) Schematic diagram of the BG computational model where two action channels are used. Continuous blue lines represent excitatory synapses, while red dashed lines represent inhibitory synapses. Single lines indicate that the synapse is represented by a scalar parameter, one per segregated channel. Thick arrows indicate that the synapses are represented by an array (2 × 2 in the present example). A double arrow line signifies that the synapses are present in both directions, although not necessarily with the same strength. D is the dopamine effect used as input for the model. (**b**) Effect of an increase in dopamine—in the figure, we assumed that neuron 1 in the motor cortex won the competition. This choice is rewarded through a dopamine peak, which excites the Go and inhibits the NoGo. The thickness and filling of the circles represent the activity of the corresponding neuron (large thickness with the filled interior is used for the maximally activated neurons), while the thickness of the connections represents the amount of information transmitted from the upstream to the downstream neurons. A reward potentiates the winning action. (**c**) Effect of a decrease in dopamine—in the figure, we assumed that neuron 2 in the motor cortex won the competition. This choice is punished through a dopamine dip, which inhibits the Go and excites the NoGo. In this condition, the winning action, initially at a high value (coded by neuron 2 in the motor cortex), is depressed and the alternative action (coded by neuron 1) may emerge, or even no response may occur. This change is illustrated through arrows on the motor cortex neurons.

**Figure 13 ijms-23-03452-f013:**
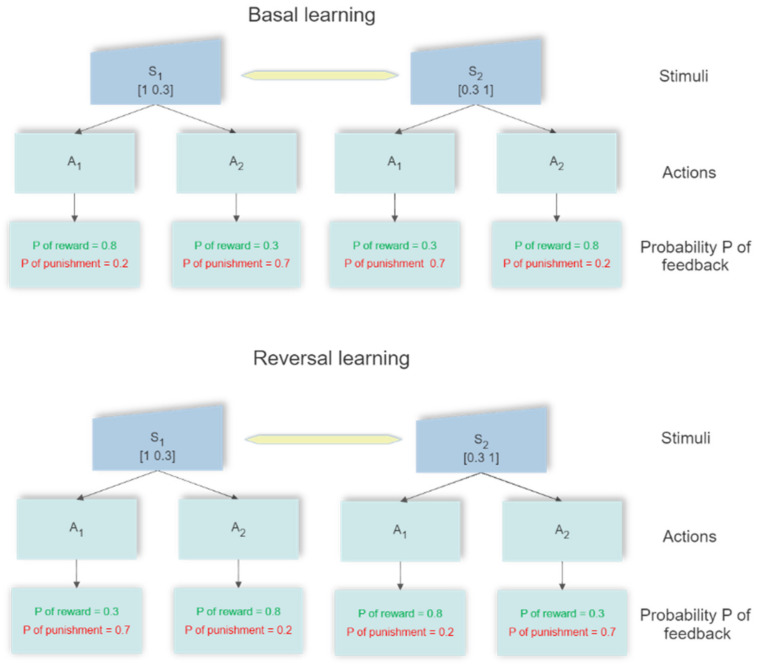
Training design of two-choice experiment. The upper panel refers to the basic learning phase: after the presentation of the stimulus S_1_, the first action (in this case A_1_) has a probability P of 0.8 to be rewarded (otherwise punished), whereas the second action A_2_ is rewarded with a probability of 0.3 (otherwise punished). When a stimulus S_2_ is given, the reward feedback probabilities are 0.3 and 0.8 for A_1_ (first action) and A_2_ (second action), respectively. The thick arrow between the stimuli S_1_ and S_2_ indicates the random permutation. It is worth noting that action A_1_ is more frequently rewarded for the stimulus S_1_, and action A_2_ for the stimulus S_2_, during basal training. The bottom panel gives a schematic representation of the reversal learning phase: when S_1_ is presented, action A_1_ receives a positive feedback with a probability of 0.3 (otherwise punished), whereas the probability of reward for the second action A_2_ is 0.8 (otherwise punished); after S_2_ presentation, A_1_ and A_2_ are rewarded with a probability of 0.8 and 0.3, respectively, (otherwise punished). During reversal learning, the second action (A_2_ for the stimulus S_1_ and A_1_ for the stimulus S_2_) refers to actions with reward probabilities as high as 0.8, i.e., the more frequently rewarded actions.

## Data Availability

Not applicable.

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
