# Peer review of "Phasic Dopamine Changes and Hebbian Mechanisms during Probabilistic Reversal Learning in Striatal Circuits: A Computational Study"

_ijms, 2022, doi:10.3390/ijms23073452_

Round 1
Reviewer 1 Report
In general, the manuscript by Schirru et al is fine. The main objection I have is that it is a bit "wordy", the real value of the manuscript, i.e., the model, is almost hidden in the text. This is especially important in the Discussion where the “value” of the model is snowed under a rather lengthy discussion, subheadings might help. Firstly, the model should be better, no “discontinuous” lines, just one line with only arrows at the end. Bigger lines should be more clearly bigger. Further, it would be useful to show the flow of information without dopamine and with dopamine next to each other, showing the changes caused by dopamine, especially as far as the output of the BG network goes. The other minor part is that it should be made clear if this model applies to humans and/or animals (e.g., rats)?
Author Response
The response to both Reviewers can be found in the enclosed file

Reviewer 2 Report
The paper proposes a neurocomputational model of the basal ganglia dopaminergic system in a probabilistic reinforcement learning environment. The authors analyze different Hebb rules and the mechanism for modulating the phasic dopamine changes. The behavior of the proposed model is interesting and provides insights into behavior deficits in neuropsychiatric or neurological disorders.
The paper is well-written and easy to follow. The literature review is comprehensive and provides enough background about the problem authors try to solve. The proposed model design is convincing though I did not check the details. As the authors promised in the manuscript, the source code will be available upon publication for reproducibility.
I am leaning towards acceptance if the authors could clarify the following questions:
1. Line 443, the authors mention 10 subjects. Are these human subjects or the proposed model running 10 times? If the latter, I found that the term subject is confusing and the authors might want to consider the term agent instead which is commonly used in reinforcement learning literature.
2. It might be interesting to look into human behaviors in these tasks and compare changes in human performance across training epochs and compare it with the proposed model.
3. How did the authors decide the probability values in each stimulus (e.g. why 0.3 in the two-choice experiment)? What is the reasoning behind it?
4. It seems that value m (line 952-955) suggests the difficulty of reversal learning. How is hyperparameter m related to biological neurons and how can m be learned? Can the value m generalize to any tasks?
Author Response

(The authors gave the same response as above.)
